# Clinical Significance and Tumor Microenvironment Characterization of a Novel Immune-Related Gene Signature in Bladder Cancer

**DOI:** 10.3390/jcm12051892

**Published:** 2023-02-27

**Authors:** Zhaohui Wang, Tao Wang, Gangfeng Wu, Lei Zhu, Jian Zhang

**Affiliations:** 1Department of Gynecology and Obstetrics, Xiangya Hospital, Central South University, Changsha 410008, China; 2Advanced Biological Screening Facility, BioQuant, Heidelberg University, 69120 Heidelberg, Germany; 3Department of Surgery, Medical Faculty Mannheim, Heidelberg University, 68167 Mannheim, Germany; 4Department of Urology, Shanghai General Hospital, Shanghai Jiaotong University School of Medicine, Shanghai 200080, China; 5Department of Urology, Shaoxing People’s Hospital, Shaoxing 312000, China; 6Junior Clinical Cooperation Unit Translational Surgical Oncology (A430), German Cancer Research Center (DKFZ), 69120 Heidelberg, Germany

**Keywords:** bladder cancer, immune-related genes, biomarker, prognosis, tumor microenvironment

## Abstract

Cancer immunotherapy plays a crucial role in bladder cancer (BC) progression. Increasing evidence has elucidated the clinicopathologic significance of the tumor microenvironment (TME) in predicting outcomes and therapeutic efficacy. This study sought to establish a comprehensive analysis of the immune-gene signature combined with TME to assist in BC prognosis. We selected sixteen immune-related genes (IRGs) after a weighted gene co-expression network and survival analysis. Enrichment analysis revealed that these IRGs were actively involved in Mitophagy and Renin secretion pathways. After multivariable COX analysis, the IRGPI comprising *NCAM1*, *CNTN1*, *PTGIS*, *ADRB3*, and *ANLN* was established to predict the overall survival of BC, which was validated in both TCGA and GSE13507 cohorts. In addition, a TME gene signature was developed for molecular and prognosis subtyping with unsupervised clustering, followed by a panoramic landscape characterization of BC. In summary, the IRGPI model developed in our study provided a valuable tool with an improved prognosis for BC.

## 1. Introduction

Bladder cancer (BC) is one of the most common malignancies of the urinary tract [1], which can be classified into two different groups based on the tumor invasion: non-muscle invasive bladder cancer (NMIBC) and muscular invasive bladder cancer (MIBC) [1,2]. While MIBC accounts for roughly 20% of BC, approximately fifty percent of MIBC patients eventually develop the disease at distant sites due to disseminated micrometastases even though they underwent radical cystectomy and bilateral pelvic lymphadenectomy [3,4]. Platinum-based perioperative chemotherapy is widely employed to treat micrometastatic disease and enhance overall survival (OS) [2,3,4]. However, the response rate to cisplatin-based agents does not exceed 50% [5]. Therefore, the prognosis of clinically localized MIBC is dismal.

The tumor microenvironment (TME) consists of various cell types, including tumor cells, immune cells, stromal cells, and extracellular matrix (ECM) [6]. There is accumulating evidence that the interaction between cancer cells and TME affects therapy responses and clinical outcomes [7]. In this context, a series of immunological checkpoint molecules associated with immune evasion is essential in regulating TME, the blockade of which offers new therapy options for patients with locally advanced and metastatic BC [8,9]. However, only a tiny fraction of patients could benefit from immune checkpoint inhibitors (ICIs) [9]. A more comprehensive and detailed characterization of TME is urgently needed to identify an improved BC stratification for better prediction of therapeutic response.

In recent years, genomic analysis has been used to identify promising prognostic factors, mine valuable therapeutic targets, and analyze the molecular mechanisms in BC. While previous works of the literature pointed out that a single biomarker is insufficient for BC prognosis prediction, the signature combining multiple genes might significantly improve prediction accuracy [10,11,12,13,14,15,16]. Given the importance of the interaction between BC and immune regulation with regard to prognosis and treatment, we sought to construct an immune gene-based signature and quantify the TME infiltration pattern for subtyping through bioinformatic analysis, which was then systematically correlated with genomic characteristics and clinicopathologic features to achieve a novel prognostic model with comprehensive landscape characterization of BC.

## 2. Materials and Methods

### 2.1. Clinical Samples and Data Acquisition

The clinical and RNA-sequencing data of BC were downloaded from The Cancer Genome Atlas (TCGA) database (https://portal.gdc.cancer.gov/projects/TCGA-BC, accessed on 5 September 2021) and GSE13507 in the Gene Expression Omnibus database (https://www.ncbi.nlm.nih.gov/geo/, accessed on 5 September 2021) [17] (Appendix A). The BC expression profiles from TCGA were composed of 411 cancer samples and 19 adjacent samples, and the RNA data from GSE13507 included 165 tumor samples. Moreover, the ImmPort database was also used to obtain a list of IRGs (https://www.immport.org/shared/home, accessed on 5 September 2021) [18].

### 2.2. Identification and Functional Enrichment Analysis of Differentially Expressed IRGs

The limma R package was used to identify differentially expressed genes (DEGs) between BC and normal tissues from TCGA (threshold: *p* < 1 × 10^−3^, |log_2_FC| > 1). Then, the differentially expressed immune-related genes (IRGs) were generated by combining them with the immune-related gene information obtained from the Immunology Database and Analysis Portal (ImmPort) database. The molecular mechanisms of the differentially expressed IRGs were further explored with the clusterProfiler function package through Gene ontology (GO) and Kyoto Encyclopedia of Genes and Genomes (KEGG) pathways enrichment analyses [19,20].

### 2.3. Weighted Gene Co-Expression Network Analysis

The weighted gene co-expression network analysis (WGCNA) has been described by [21] as a system of the biology approach that uses gene expression data to construct scale-free networks. By calculating the absolute value of the correlation between the expression levels of transcripts, a co-expression similarity matrix was constructed. Next, the adjacency matrix was created and converted into a topological overlap matrix when the power of β = 5. Afterward, we constructed a hierarchical clustering dendrogram of the 1-TOM (Topological Overlap Matrix) and used the dynamic hybrid cut method to identify co-expression gene modules with a module minimum size cutoff of 30. By setting the minimum height to 0.25, the modules with highly correlated genes were merged. To uncover functional modules closely related to BC in the co-expression network, we calculated the module–trait associations between modules and clinical trait information.

### 2.4. Identification and Molecular Characteristics of Hub IRGs

According to the significantly related co-expression gene set revealed by WGCNA, the network was constructed by selecting edges (weight > 0.2), followed by analyzing the degree in the network. The top 202 nodes (degree > 2) were selected as the hub genes of the network. The final hub IRGs were screened out through survival analysis after finding the best cutoff with the maxstat R package. In order to study the genome-wide characterization of hub IRGs, we also downloaded the gene mutation information of hub IRGs from the cbioportal (http://www.cbioportal.org/, accessed on 5 September 2021) database. The regulatory network based on hub IRGs was established for TF and ncRNA (miRNA, lncRNA), with RAID 2.0 (http://www.rna-society.org/mgdr/home, accessed on 5 September 2021), TRRUST v2 (https://www.grnpedia.org/trrust/, accessed on 5 September 2021) and String database as the background. It was used to explore the mutual regulation between hub IRGs and transcription factor (TF) and ncRNA (miRNA, lncRNA).

### 2.5. Development of Immune-Related Gene Prognosis Index Model and Evaluation of its Prognostic Efficacy

To obtain the significantly related genes to BC, we performed the multivariate Cox regression analysis. The immune-related gene prognosis index (IRGPI) was generated by considering the expression of each significantly related gene and its weight. The formula is shown below (W represents the weight, and E represents the expression value of the gene). Then, the prognostic efficacy of the index was evaluated by survival analysis and the receiver operating characteristic (ROC) curve by analyzing TCGA data with the survivalROC R package and further verified with GSE13507 from the GEO database.
index=∑w×E

### 2.6. Unsupervised Clustering Molecular Typing of the Tumor Microenvironment

With the default signature matrix at 1000 permutations, the BC gene expression data from TCGA was imported into CIBERSORT (https://cibersort.stanford.edu/, accessed on 5 September 2021) to obtain the relative proportions of 22 types of infiltrating immune cells, such as CD8^+^T cells, CD4^+^T cells, and B cells [22]. The best K will be determined from a combination of elbows and gaps, i.e., the point where Wk drops the fastest and the K that corresponds to the largest gap [23]. The Consensus ClusterPlus R package was used for patient classification to obtain TMEcluster (kmeans, euclidean, and ward. D), with the repetition of 1000 times to ensure stable results [23]. The survival data were applied to evaluate whether this classification was related to survival.

### 2.7. Prediction of the Response to Immunotherapy and Chemotherapy

To evaluate the clinical response of ICIs, we used the TIDE (http://tide.dfci.harvard.edu/, accessed on 5 September 2021) tool [24]. Higher tumor TIDE prediction scores indicated a poorer immune checkpoint therapeutic effect. To observe the sensitivity to the anti-cancer drugs cisplatin and Gemcitabine, we explored the pRRophetic R package to predict the IC50 drug sensitivity in the BC subtype (IRGPI high/low) [25].

### 2.8. Statistical Analysis

Univariate and multivariate Cox regression analyses were performed using the survival R package. The areas under the ROC were calculated with the survival ROC R package to validate the performance of the prognostic index. An independent *t*-test was selected to identify the differences among clinical parameters; *p*-values below 0.05 were regarded as statistically significant. R software was used for all statistical analyses (version 4.0.2; https//www.Rproject.org/, accessed on 5 September 2021).

## 3. Results

### 3.1. Differentially-Expressed Gene Screening and Immune-Related Genes Enrichment Analysis

The DEGs between BC and paracancerous samples were screened out from the analysis of the TCGA cohort (Figure 1A,C). Using |log_2_FC| > 1 and *p* < 1 × 10^−3^ as the threshold, 1563 genes were identified, including 692 upregulated and 714 downregulated genes. Then, the target differentially-expressed IRGs (N = 419, 137 higher expressions versus 282 lower expressions) were generated by combining immune-related gene information obtained from the ImmPort database (Figure 1B,D, Appendix A). GO and KEGG pathways enrichment analyses were performed to clarify the biological processes and pathways related to IRGs, revealing that these genes were mainly involved in ECM-receptor interaction, Focal adhesion, and PI3K-Akt signaling pathway (Figure 1E,F, Appendix A).

### 3.2. Weighted Gene Co-Expression Network Establishment and Key Modules Identification

According to the protein interaction information in the STRING database, genes interacting with 419 differentially expressed IRGs were also included in the reference range, resulting in a total of 3662 reference genes (Appendix A). Among them, 3611 genes had expression values in the TCGA data, which were used as the input dataset for WGCNA to reveal the functional clusters in BC patients. With the “WGCNA” package in R, 12 modules were generated (Figure 2A). Then, the heatmap of module–trait relationships was constructed to evaluate the association with clinical parameters. The Pearson correlation coefficient between each module’s eigengenes (ME) and the sample characteristics was calculated. The results of the relationships between the modules and the traits are shown in Figure 2B. Figure 2C,D shows that the blue and brown modules were the most relevant to BC.

The functional enrichment analysis was performed with the genes of the blue and brown modules. While the brown module was mainly enriched in calcium signaling, vascular smooth muscle contraction, and hypertrophic cardiomyopathy (Figure 2E), the blue module was mainly associated with the cell cycle, proteasome, and DNA replication (Figure 2F, Appendix A).

### 3.3. Identification of Hub Immune-Related Genes

According to the results of WGCNA, the brown and blue modules were selected for subsequent analysis. The network was constructed by selecting edges with a weight > 0.2 (Figure 2D, Appendix A), and the nodes in the network with a degree greater than 2 (Appendix A) were considered as the hub genes. Accordingly, 202 genes were identified as the hub genes, among which the IRGs were selected as hub IRGs (N = 52). After survival analysis, 16 hub IRGs significantly related to survival time were obtained as the final hub IRGs. The four most significant IRGs (BOC, NCAM1, PTGIS, and ITGA7) are shown in the figure below (Figure 3A–D, Appendix A). Then, the multivariate Cox regression analysis was conducted to analyze the correlation between hub IRGs and BC after adjusting for clinical information, including age, sex, grade, AJCC stage, pT stage, pN stage, and pM stage. The figure below shows that NCAM1, CNTN1, PTGIS, ADRB3, and ANLN were significantly related to BC (Figure 3E).

### 3.4. Characterization of the Regulatory Network, including Hub IRGs and TF and ncRNA (miRNA, lncRNA)

The cbioportal database was used to analyze the mutation statuses of the hub IRGs (Appendix A). The mutation frequency of NCAM1, CNTN1, PTGIS, ADRB3, and ANLN was 2.7%, 4%, 2.2%, 11%, and 6%, respectively.

With RAID 2.0, TRRUST v2, and STRING database as the background, the regulatory network incorporating TF and ncRNA (miRNA, lncRNA) was established based on the hub IRGs. In total, 21 hub/miRNA, 12 hub/ncRNA, and 21 hub/TF interaction pairs were generated. After de-redundancy, there were 52 interaction pairs and 40 genes (Figure 4A, Appendix A). As shown in Figure 4A, round nodes represented TF, squares represented mRNA, diamonds represented lncRNA, and triangles represented miRNA.

The genes in the regulatory network were then subjected to KEGG enrichment analysis using Cytoscape’s ClueGO plug-in. As shown in Figure 4B, Mitophagy and Renin secretion were the mainly enriched pathways (Appendix A).

### 3.5. Establishment of an IRGPI Model and Evaluation of its Prognostic Efficacy

According to the above findings, NCAM1, CNTN1, PTGIS, ADRB3, and ANLN were significantly correlated with BC. The prognostic model for predicting OS was calculated based on the following formula: IRGPI = NCAM1 × 0.523 + CNTN1 × 0.245 + PTGIS × 0.218 + ADRB3 × (−1.035) + ANLN × 0.250. After combining with clinical information, IRGPI was significantly related to OS (*p* < 0.0001, Figure 5A, Appendix A). Although the area under the curve (AUC) was 0.562 in the TCGA cohort, the IRGPI model was further verified in the GSE13507 cohort, with the AUC reaching 0.644 (Figure 5B–D, Appendix A).

### 3.6. Unsupervised Hierarchical Clustering of Molecular Subtypes and Construction of a Comprehensive BC Landscape

The data of cancer samples from the TCGA cohort (N = 408) were imported into the CIBERSORT analytical tool and were repeated 1000 times to obtain the different immune cell ratios of all cancer samples (Appendix A). Combining elbow to evaluate the number of best categories, K declined when K = 3 (Figure 6A). Combined with clinical data, this classification (K = 3, 4) was significantly related to survival (K = 3: *p* = 0.012; K = 4: *p* = 1 × 10^−4^; Figure 6B,C), so K = 4 was selected. As shown in Figure 6D, the prognosis for Cluster 1 and Cluster 2 was better, while Cluster 3 and Cluster 4 were poor. A comprehensive BC landscape was constructed by combining the immune cell component ratios, the clinical parameters, the unsupervised clustering results of the TME, and the IRGPI grouping of the tumor samples to describe the molecular and clinical characteristics classified by different subtypes (Figure 6E, Appendix A).

### 3.7. Prediction of the Response to Immunotherapy and Chemotherapy

The TIDE tool was explored to assess the potential efficacy of immunotherapy (Appendix A). A higher score indicated a poorer immunotherapy efficacy. According to the grouping information of IRGPI in this study, there is no significant difference in the TIDE score between the high and low groups (Wilcoxon-test, *p* = 0.8; Figure 7A). However, the expression of CD274 (PD-L1), myeloid-derived suppressor cells (MDSCs), and cancer-associated fibroblasts (CAFs) showed significant differences between the two groups (Figure 7B–D). In addition, the drug resistance to cisplatin and Gemcitabine in different IRGPI groups was predicted with the pRRophetic package. Compared with the IRGPI high group, the IC50 of the IRGPI low group was significantly higher, indicating a worse effect of platinum-based chemotherapy in the IRGPI low group (Figure 7E,F; Appendix A).

## 4. Discussion

Immunotherapy has revolutionized clinical cancer treatment and shown a remarkable, durable response in BC [26,27,28]. Recent research has revealed that the signatures of IRGs are effective prognostic biomarkers in various cancers, including BC [10,11,12,13,14,15,16,29,30,31,32]. In the current study, we used the TCGA RNA-seq data to screen the DEGs between tumor and normal samples. After focusing on immune-related DEGs by combining the ImmPort database, WGCNA was performed to find hub IRGs. Furthermore, multivariate Cox regression analysis was used to construct an IRGPI model for prognostic indicators, followed by an independent validation of this panel in one GEO database (GSE13507). Moreover, the landscape of BC and drug resistance were further depicted, which showed that the patients in IGRPI high group displayed a worse OS but better possible response to cisplatinum and Gemcitabine, indicating that IRGPI was a valid prediction and prognosis model for BC.

Activation of immune checkpoints leads to cancer-induced immune suppression and plays a key role in cancer progression [33,34]. ICIs that block the PD-1/PD-L1 pathway can lead to durable remissions in patients with various cancers [35]. Furthermore, MDSCs and CAFs were the important reasons for restricting T cell infiltration in tumors [24]. Our data showed that the expression levels of PD-L1, MDSCs, and CAFs were significantly higher in the IRGPI score high group. However, our model could not predict the effects of immunotherapy by analyzing with TIDE tools, suggesting other potential mechanisms being involved in immunotherapy resistance of BC patients in the IRGPI high group, which warranted further investigation.

In our study, ADRB3 was uncovered as crucial in this IRGPI model and acted as a protective factor in BC patients’ survival. Previous studies have reported that ADRB3 belonged to the β-adrenergic receptor family [36]. The Trp64Arg polymorphism in ADRB3 increased the risk of endometrial cancer but was associated with decreased susceptibility for breast cancer [37]. There are no reports concerning the function and molecular mechanisms of NCAM1, CNTN1, and ADRB3 in BC. However, the other two of them (ANLN and PTGIS) have been studied. Higher ANLN transcript levels were associated with worse OS and disease-specific survival (DSS) of a BC cohort and may be an independent predictor for progression-free survival (PFS) of BC [38]. Knockdown of ANLN could suppress the proliferation, invasion, and migration, and lead to G2/M phase arrest of BC cells, indicating a promising role for ANLN as the prognostic biomarker for BC patients [39]. PTGIS was reported to be downregulated in BC and transcriptionally inhibited by HIF-1α via binding to the promoter region or promoting its DNA methylation [40]. Moreover, PTGIS has been identified as a key gene in another prognostic model, suggesting its crucial role in the progression of BC [16]. However, owing to the differences in modeling methods and the inherent heterogeneity of BC, no other genes were found in common between this and other studies [10,11,12,13,14,15].

To explore the potential molecular mechanisms of the genes in our model in promoting the progression of BC, a TF-mediated network was constructed to unveil significant TFs involved in regulating hub IRGs. After KEGG pathway enrichment analysis, the hub IRGs were significantly enriched in the pathway of renin secretion. Previous studies have confirmed that renin and ADRB3 were related to the renin-angiotensin system (RAS) [36], which is vital in regulating blood pressure and cardiovascular homeostasis. Recent advances suggested that local RAS was related to an immunosuppressive TME caused by myeloid cells and CAFs, which contributed to effector T- cell dysfunction, apoptosis, and failure to infiltrate deep into the tumor [41,42]. Pharmacological inhibition of RAS was related to improved outcomes in patients with BC [43,44,45]. In addition, as a component of RAS, Juillerat-Jeanneret et al. revealed that renin was significantly associated with the proliferation and survival of glioblastoma cells, independent of the action of angiotensin peptides on their cognate receptors [46]. As an endogenous 7–amino acid peptide hormone of the RAS, Katherine et al. demonstrated that Angiotensin-(1-7) [Ang-(1-7) inhibited the growth of CAFs and reduced fibrosis in the TME [47]. Another study developed the nanoconjugates of angiotensin receptor blockers (ARBs) that preferentially accumulated and acted in tumors. These ARBs may reprogram CAFs to activate T-cell-mediated immune response and improve the efficacy of ICIs in mice with primary and metastatic breast cancer [48]. All the above findings suggested that the hub IRGs in our model may promote the progression of BC by increasing the infiltration of CAFs via the pathway of renin secretion and may serve as a robust prognostic and predictive tool.

As a prominent component in the tumor stroma, CAFs are spindle-shaped cells that play a vital role in constructing and remodeling the ECM structure and reprogramming TME [49,50]. Several studies have linked CAFs to BC progression and poor prognosis [51,52,53,54] and had the potential utility in anti-cancer immunotherapy [49,55,56]. Zhuang et al. reported that CAFs induced the epithelial–mesenchymal transition and invasion of BC through the TGFβ1-ZEB2NAT-ZEB2 axis [54]. By calculating the CAFs infiltration score, Liu et al. demonstrated that high infiltration of CAFs was associated with poor OS and cancer-specific survival (CSS) in BC [52]. As ICIs can induce favorable responses in cancer therapy [57,58,59], the expression of CAFs and PD-L1 was significantly higher in the IRGPI score high group in our study. However, our model could not predict the effects of immunotherapy by analyzing with TIDE tools, indicating the potential role of CAFs in promoting immune evasion of BC except for the PD-1/PD-L1 pathway. As immunotherapeutic approaches combining “anti-CAF” and “anti-PD-L1” have exhibited some promising results [50]; this combined therapy may contribute to a potential strategy for BC patients with high IRGPI scores.

In addition, TME has been reported to play a significant role in the pathogenesis of BC. Yeh et al. explored the role of estrogen receptors-α (ERα) in the TME and their impact on BC’s progression [60]. They found that CAFs may secret CCL1 to enhance the invasion of BC via upregulating the expression of ERα, which indicated a promising therapeutic target in BC treatment [60]. The prognostic signature developed from TME may serve as a novel predictive biomarker and improve the prognosis of BC patients receiving ICIs [23]. By combining with the IGRPI model, we developed the TME gene signature to predict survival and evaluate the vast landscape of interactions between the clinical characteristics of BC and infiltrating immune cells. The characterization of the TMEscore subtype associated with potential biological pathways warranted further exploration.

There were some limitations in this study. Firstly, while the clinical effects of chemotherapy were predicted with the pRRophetic package, cohorts with complete follow-up data are further needed to validate the predicted results. Furthermore, the landscape of BC characterized by the IRGPI model and TME signature might be a useful predictive tool to identify patients who might benefit from immunotherapy, which warranted further investigation. In addition, this was a retrospective study, and further prospective studies are needed to validate our findings.

## Figures and Tables

**Figure 1 jcm-12-01892-f001:**
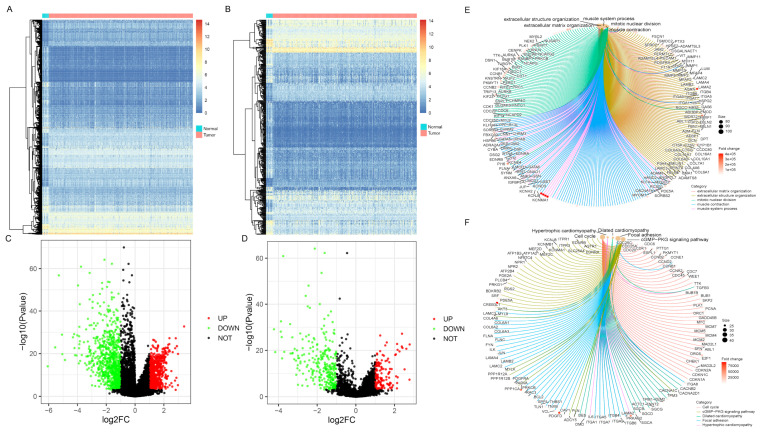
Differentially-expressed gene screening and enrichment analysis of immune-related genes (IRGs). Heatmaps represented differentially expressed genes or IRGs between BC and normal samples, respectively (**A**,**B**). Green dots in volcano plot demonstrated down-regulated genes or IRGs, and red dots demonstrated up-regulated genes or IRGs, respectively (**C**,**D**). The enrichment analysis of biological processes and pathways were shown via Gene Ontology (GO) and Kyoto Encyclopedia of Genes and Genomes (KEGG) (**E**,**F**).

**Figure 2 jcm-12-01892-f002:**
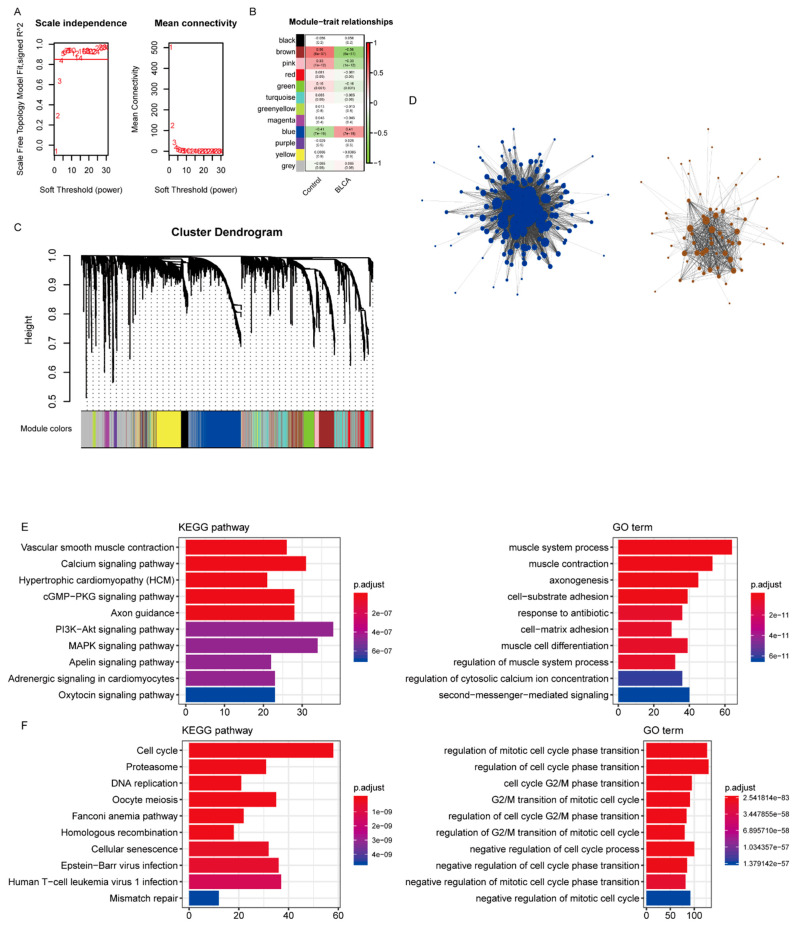
Identification of modules associated with bladder cancer (BC). (**A**) Analysis of the scale-free fit index and average connectivity of the 1–20 soft threshold power (**B**) In order to ensure that the network is a scale-free network, the optimal β = 5 was selected. B. Module–trait relationships. (**C**) Genes were grouped into various modules by hierarchical clustering. Each module has been color coded. (**D**) The network construction with the significantly correlated brown and blue modules. (**E**,**F**) The functional enrichment analysis of the genes in the brown and blue modules.

**Figure 3 jcm-12-01892-f003:**
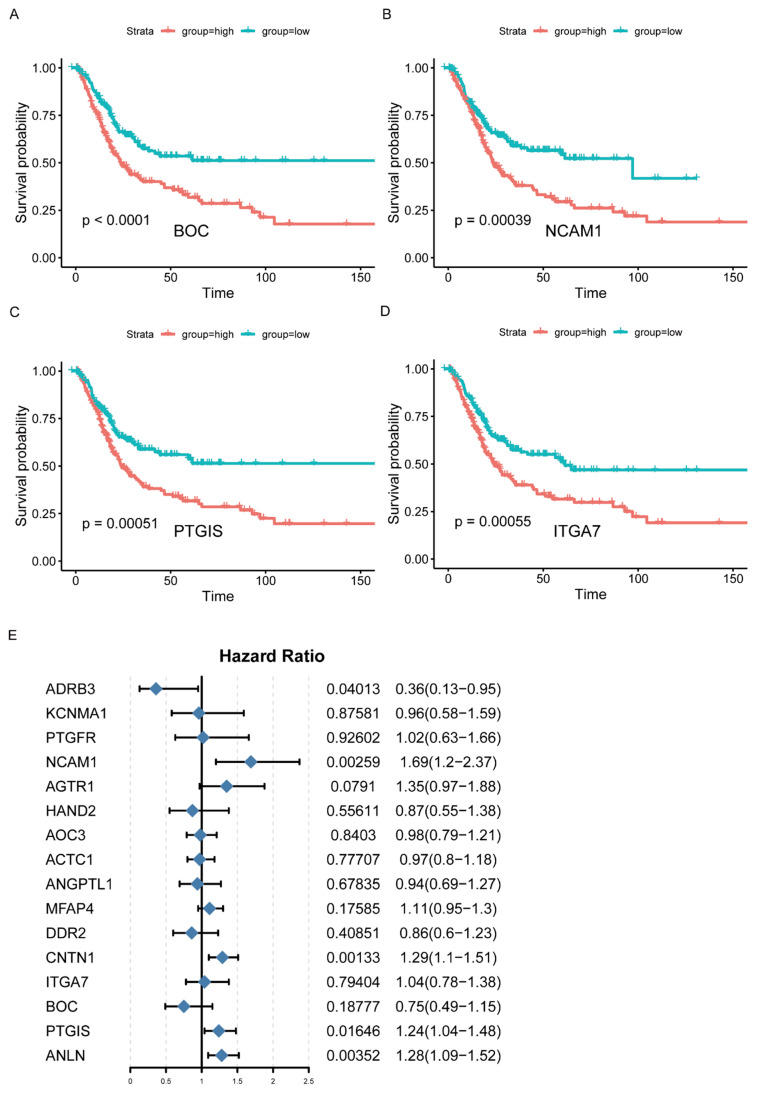
Identification of hub IRGs associated with BC. (**A**–**D**) Kaplan–Meier survival curves of hub IRGs in BC patients from TCGA database (top 4). (**E**) Multivariate Cox regression analysis of hub IRGs.

**Figure 4 jcm-12-01892-f004:**
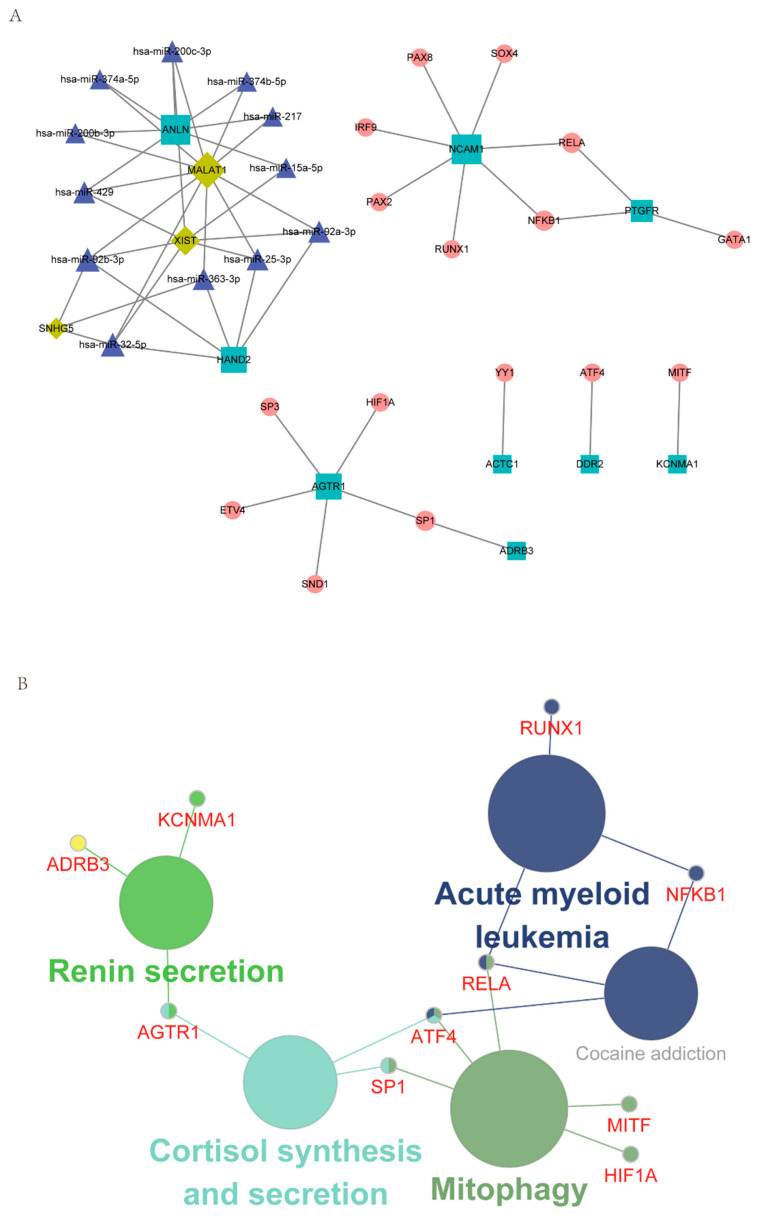
Characterization of the regulatory network, including hub IRGs and TF and ncRNA. (**A**) The regulatory network of hub IRGs and TF and ncRNA. (**B**) KEGG enrichment analysis of genes in the integrated regulatory network.

**Figure 5 jcm-12-01892-f005:**
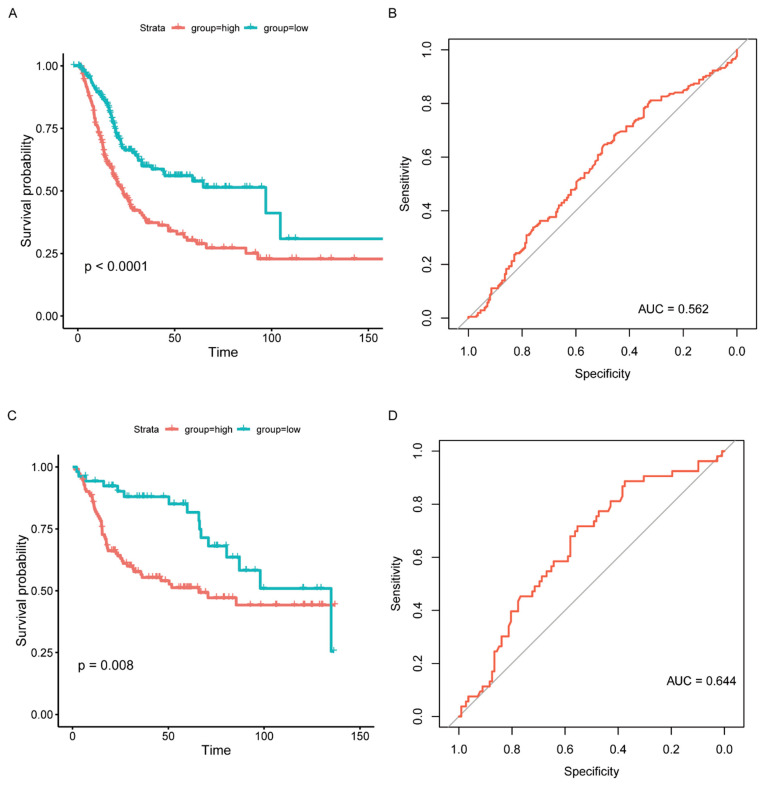
Prognostic model validation and survival outcomes of different risk groups. (**A**) Kaplan–Meier curves of different risk groups in the TCGA cohort. (**B**) ROC curve of the prognosis model (TCGA). (**C**) Kaplan–Meier curves of different risk groups in the GSE13507 cohort. (**D**) ROC curve of the prognosis model (GSE13507).

**Figure 6 jcm-12-01892-f006:**
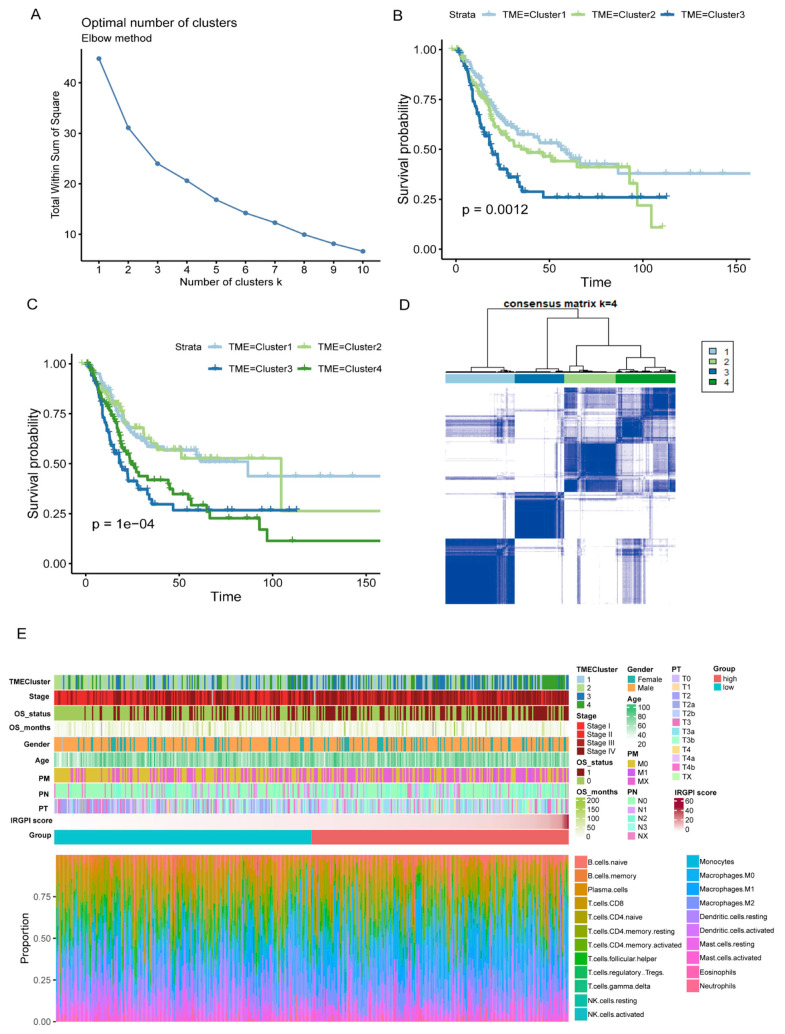
Characterization of TME subtypes and landscape of BC. (**A**) Elbow method to find the best cluster K value. (**B**,**C**) Kaplan–Meier survival curves from TCGA cohort with the TME infiltration classes (K = 3 and 4). (**D**) Consensus matrix (K = 4). (**E**) Construction of BC landscape based on the prognostic index, the unsupervised clustering of TME, the infiltration ratios of immune cells, and clinical information.

**Figure 7 jcm-12-01892-f007:**
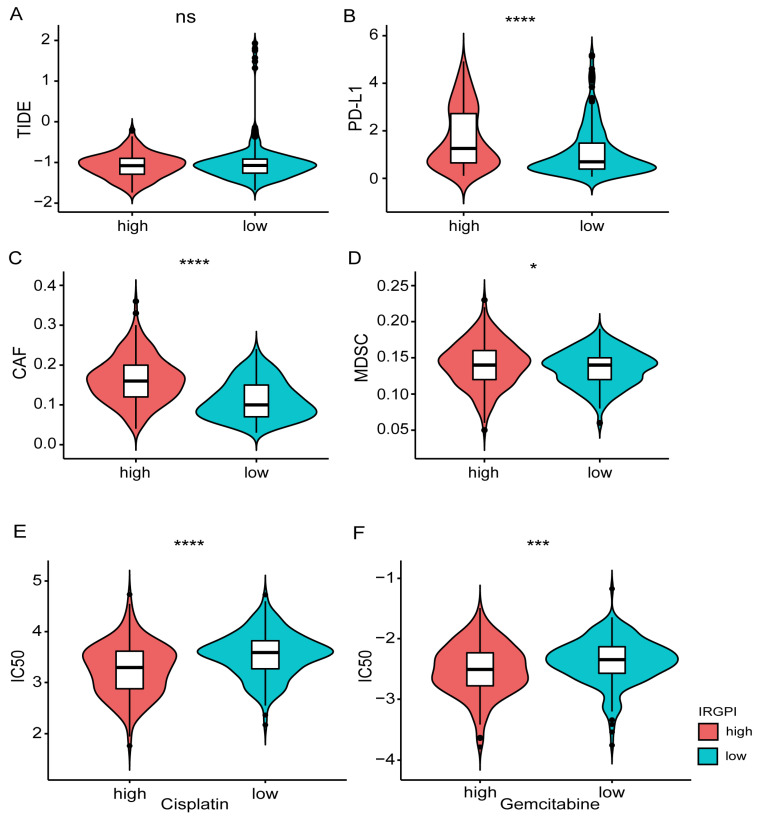
Prediction of the response to immunotherapy and platinum-based chemotherapy. (**A**) Comparison of immune checkpoint therapy efficacy based on the TIDE scores between different IRGPI groups. (**B**–**D**) Comparison of the expression of PD-L1, MDSC, CAFs between different IRGPI groups. (**E**,**F**) Comparison of the drug resistance (Cisplatin, Gemcitabine) between different IRGPI groups. * *p* < 0.05, *** *p* < 0.001, **** *p* < 0.0001.

## Data Availability

The RNA-sequencing data that support the findings of this study can be downloaded from the UCSC Xena website (https://gdc.xenahubs.net, accessed on 5 September 2021) and the Gene Expression Omnibus database (GEO, https://www.ncbi.nlm.nih.gov/geo/, accessed on 5 September 2021), respectively.

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
