# Peer review of "Clinical Significance and Tumor Microenvironment Characterization of a Novel Immune-Related Gene Signature in Bladder Cancer"

_jcm, 2023, doi:10.3390/jcm12051892_

Round 1

Reviewer 1 Report

The manuscript is easy to read, well prepared, with the use of multiple modern data mining tools, and extensive background literature searches. 

Items for consideration:

1- According to citations 10-16, it seems that there are several papers addressing predictive signatures (Line 59).  There is however no further reference to these papers in the Discussion to explain differences in the genes of interest.

2- Area under the curve of 0.562 is not “moderate” predictive value, it is weak (close to 0.50), but the validation study cohort based on GSE13507 showed a more promising predictive value (AUC=0.644) Lines 231 and 237). Prospective clinical trials may show better results, as suggested in Line 357. Application of this model to other GSE data cohorts could enhance the value of the manuscript and increase interest in applying the model to prospective clinical trials.

3- In clinical practice, decisions regarding management vary for stage T1 and stages T2 and above. Would the new predictive tool be more useful in one of these two scenarios than in the other? 

4- Model was useful for predicting response to therapy, but data results are only shown in a Supplement (Lines 266-271).  Considering the goals of the manuscript, at least some of the data could be presented in the main body of Results section.

Reviewer 2 Report

In this study, the authors investigated immune based genes from Urothelial bladder cancers along with the tumor microenvironment in order to create a prognostic model that would be able to improved bladder cancer patient stratification according to overall survival and response to systemic therapy.

The authors based their bladder cancer data on well-known cancer databases (TCGA, GEO and ImmPort) and present a very sophisticated and thorough process for the identification of specific immune related genes that were eventually used in the prognostic models.

These steps included:

       i.          Gene sorting – bladder cancer genes and paracancerous genes were subclassified into immune related genes based on information from the ImmPort database. Those genes were further classified according to their involvement in ECM interaction and specific signaling pathways.

     ii.          Co-Expression networks – based on protein interactions, more genes that interact in IRG expression were divided into 12 modules. Two modules were selected based on the highest module-trait relationships.

   iii.          The clinical implication of these selected genes were finally presented in Figure3 where Kaplan-Meier survival curves identified 16 IRGs that were significantly associated with survival probability, of which eventually the 4 who mostly correlated with disease survival were identified (BOC, NCAM1, PTGIS, and ITGA7).

    iv.          Finally, based on Forest plots, five more genes (NCAM1, CNTN1, PTGIS, ADRB3, and ANLN) were significantly related to bladder cancer.

     v.          Based on the information above a prognostic model was established. This model manages to stratify patients into high or low survival probability with AUC of 0.562 ( p<0.0001) and AUC of 0.644 (p<0.008). The authors confirm validation of these predictive models.

    vi.          Immune cell ratios from the tumor microenvironment were used to predictive clusters of survival probability. Two TME clusters (1 and 2) were found to have a better survival prediction.

  vii.          Response to treatment was also evaluated where the TIDE score managed to predict response to platinum-based chemotherapy but was not able to predict the response to Immunotherapy.

The authors present a timely topic that has important clinical implications and reflects the future of prognostic models in bladder cancer. Despite its retrospective nature, the study was well designed and shows some interesting results.

I do believe that this study may interest the readers of Journal of Clinical Medicine, however I do have some comments:

·       Clinical information is lacking (stage, grade, prior Tx…). A table 1 could be helpful.

·       The methodology in this study is presented in a way that could be overwhelming to the average reader of Journal of Clinical Medicine.
